# Voices of the vulnerable: Exploring the livelihood strategies, coping mechanisms and their impact on food insecurity, health and access to health care among Syrian refugees in the Beqaa region of Lebanon

Dana Nabulsi[1], Hussein Ismail[1], Fida Abou Hassan[1], Lea Sacca[1], Gladys Honein-AbouHaidar[1,2]*, Lamis Jomaa[1,3]*

1 Refugee Health Program, Global Health Institute, American University of Beirut, Beirut, Lebanon, 2 Hariri School of Nursing, American University of Beirut, Beirut, Lebanon, 3 Department of Nutrition and Food Sciences, Faculty of Agricultural and Food Sciences, American University of Beirut, Beirut, Lebanon

☯ These authors contributed equally to this work.
* gh30@aub.edu.lb (GH); lj18@aub.edu.lb (LJ)

**Data Availability Statement:** Data cannot be shared publicly because of confidentiality and

## Abstract

Lebanon has approximately one million Syrian refugees (SR) registered with the United Nations High Commission on Refugees (UNHCR) and an unknown number of unregistered SR, who cannot benefit from formal assistance. This study aimed to examine the livelihoods, coping strategies, and access to healthcare among SR based on registration status and accompanying formal assistance. A mixed-method approach with more emphasis on the qualitative design was adopted. A purposive convenient sampling approach was used to recruit SR from informal tented settlements (ITS) in the Beqaa region in Lebanon. Data collection included 19 focus group discussions (FGDs) that were conducted with participants, who were further divided into three groups: registered refugees with assistance, registered without assistance and unregistered. Twelve in-depth interviews were conducted with key informants from humanitarian organizations. All interviews and FGDs were audio recorded, transcribed, and thematically analyzed. SR were highly dependent on formal assistance when received, albeit being insufficient. Regardless of registration status, refugees resorted to informal livelihood strategies, including informal employment, child labor, early marriage, and accruing debt. Poor living conditions and food insecurity were reported among all SR. Limited healthcare access and high out-of-pocket costs led to limited use of antenatal care services, prioritizing life-threatening conditions, and resorting to alternative sources of healthcare. Severity of these conditions and their adverse health consequences were especially pronounced among unregistered refugees. Our findings shed light on the economic and health disparities among marginalized SR, with the lack of registration and formal assistance increasing their vulnerability. More tailored and sustainable humanitarian programs are needed to target the most vulnerable and hard-to-reach groups.

privacy concerns of study participants who are a particularly vulnerable and 'hidden' population group with security and safety concerns. The American University of Beirut Institutional Review Board (IRB) approved the study, and the consent document that the IRB approved assured participants that their data would not be shared beyond the research team and as aggregated data in publications. The de-identified data for the study may be made available to investigators who contact AUB in accordance with institutional policies. Please note that AUB policies require AUB investigators to retain custody of research data, unless Non-Disclosure Agreements (NDA) have been signed prospectively with investigators/ collaborators in other institutions. You can also contact AUB IRB office (irb@aub.edu.lb) for any additional inquiries related to human subjects' data for research purposes.

**Funding:** The authors received no specific funding for this work.

**Competing interests:** The authors have declared that no competing interests exist.

## Introduction

Today, the Syrian conflict remains one the largest humanitarian disasters, resulting in massive destruction, loss of lives, and the displacement of over 5 million individuals seeking refuge in neighboring countries, mainly Jordan, Lebanon and Turkey [1]. Lebanon, a small middle-income country of six million people, has the largest concentration of refugees per capita and the fourth largest refugee population in the world [2–4]. Historically, Lebanon has been a host country for refugees who have fled wars and persecution in their home countries, including refugees from Palestine with the 1948 Palestinian exodus, from Iraq post the Persian-Gulf war in the 1990s and the 2003 invasion, as well as forcibly displaced individuals from Syria with the onset of the war in 2011 [5–7]. As of July 2020, the Government of Lebanon (GoL) estimated that the country hosts close to 1.5 million Syrian forcibly displaced individuals (refugees), of which around one million were either registered or recorded with the United Nations High Commissioner for Refugees (UNHCR) [8]. Of note, as of May 2015, and in response to the GoL decision, UNHCR suspended registrations of Syrian refugees. In the meantime, UNHCR has been "recording" rather than "registering" individual refugees where basic personal data of newcomers are being recorded, taking into consideration protection and assistance. In parallel, an unidentified number of refugees have never been registered nor recorded and are referred to as 'unregistered'.

To date, significant efforts have been exerted by the GoL with the support of the UNHCR, the World Food Programme (WFP) and other non-governmental, humanitarian organizations to assist Syrian refugees in Lebanon. Currently, several cash-based assistance modalities exist in the country, including the UNHCR multi-purpose cash assistance (MPC) modality that provides the most economically vulnerable registered refugees with unconditional 175 US dollars (USD) per household per month to meet their basic needs [9]. In addition, WFP provides a monthly unconditional and unrestricted transfer of 27 USD per person per month to improve the Syrian refugees' access to food [10]. As for access to health care, the UNHCR subsidizes access to primary healthcare centers for a co-payment of 2–3 USD and covers 75–90% of hospital fees for deliveries and life-saving emergencies for all registered refugees, regardless of vulnerability or assistance status [11,12]. However, these cash assistance modalities and subsidies exclude the unregistered Syrian refugees, thus the latter are expected to be more vulnerable than others due to lack of support [13].

Evidence exists that despite the substantial efforts exerted by local and international actors over the past decade, Syrian refugees registered (or recorded) with the UNHCR in Lebanon still face tremendous challenges that affect their food security, livelihoods and access to healthcare among other basic services [3,14,15]. Moreover, due to the fragmented and privatized nature of the healthcare system in Lebanon, Syrian refugees face multiple barriers to access healthcare, such as high out-of-pocket costs and transportation difficulties [16]. Syrian refugees also continue to face heighted social tension with the host community due to competition over limited livelihood and job opportunities in a resource-strained, small country like Lebanon [17,18]. These conditions have further exacerbated the living conditions of refugees and exposed them to different forms of discrimination, prejudice and scapegoating by the Lebanese host communities [17,18]. With the Syrian refugee crisis well into its 10th year [3] and the insufficient humanitarian funding to meet growing demands, it is anticipated that the refugees' situation may become more precarious amidst the recent political and financial crises that the country have been witnessing along with the ongoing COVID-19 pandemic [19–21].

As a consequence of insufficient assistance as well as limited accessibility of services and sources of livelihoods, Syrian refugees opt to adopt various coping mechanisms in order to cover daily basic needs [3]. Coping mechanisms are the actions performed by individuals and

households to solve any issue that they perceive as problematic [22], and the scientific literature is rich with examples of different coping mechanisms and strategies adopted by displaced individuals and refugees under stressful situations [23]. Some of these coping strategies focus on the emotional wellbeing of affected individuals, such as seeking social support and using religion and faith to overcome negative feelings and emotions [24,25], while other coping mechanisms adopted by refugees include food and non-food related mechanisms to overcome food shortages and economic constraints [26–28]. In Lebanon, evidence suggests that registered (and recorded) Syrian refugees are adopting various coping strategies to deal with poverty, food insecurity and the daily life and economic stressors. These strategies include food-related coping mechanisms such as eating cheaper foods or going days without eating and other non-food related coping mechanisms such as reducing expenditure on health and education [13]. On the other hand, limited evidence exists as to the livelihood and coping strategies adopted by unregistered refugees in Lebanon and in similar contexts in the region [29,30]. Knowing that unregistered refugees tend to reside in their host countries illegally, they represent a 'hidden' population group that is not included in any formal surveys or census yet may be the most vulnerable to discrimination and exploitation [31]. The term vulnerability, as defined in the present study, refers to the ability of a refugee household to manage and cope with the impact of the emergency or crisis [32].

Thus, an in-depth understanding of the livelihoods and coping mechanisms adopted by all Syrian refugees and a comparison by registration status are much needed. This study aimed to explore the livelihoods, coping strategies and access to health care among Syrian refugees in the Beqaa region in Lebanon based on their registration status (registered vs. unregistered) and accompanying formal assistance. In this paper, a mixed methods approach with more emphasis on the qualitative design was adopted to address the research objectives. Further details were provided in the subsequent section. Findings from the present study can assist humanitarian workers, public health professionals, and decision makers in improving their understanding of the livelihood strategies and challenges faced by Syrian refugees in Lebanon, particularly the 'invisible' group of unregistered and unrecorded refugees. In addition, these findings can help policy makers and practitioners in developing new programs and services that are sensitive to the needs of hard-to-reach groups of refugees to improve their living conditions and their access to health care and other basic services.

## Materials and methods

### Study design and setting

This study was based on a mixed method approach with more emphasis on the qualitative design. The research team adopted an exploratory descriptive qualitative design including focus group discussions (FGDs) supplemented with a short questionnaire with Syrian refugees and in-depth interviews with key stakeholders. The study was carried out in the Beqaa Valley in Lebanon, specifically in the Baalbek-Hermel governorate. This specific region was chosen for various reasons. First, this region is home for more than 344,000 registered refugees as per the latest UNHCR estimates [33], equivalent to 37% of the total Syrian refugee population registered in Lebanon [34]. Most Syrian refugees are residing in overcrowded non-permanent shelters and in informal tented settlements (ITS) [10]. In addition, the Baalbek-Hermel region is close to the Syrian borders and it is believed that an unknown number of unregistered refugees reside there due to its proximity to the border. Unregistered refugees represent a particularly 'hard to reach' group that lacks official status in the country and are more likely to reside in ITS located in remote rural areas in the country.

The present study was conducted according to the guidelines laid down in the Declaration of Helsinki, and the study protocol and procedures were approved by the Social and Behavioral Sciences Institutional Research Board at the American University of Beirut (Protocol number SBS-2018-0266).

## Research team

The research team consisted of experts in mixed methods approaches (LJ, GHA) and moderators (D.N, H.I and F.AH) who were well-trained on how to conduct qualitative research and how to effectively and ethically facilitate FGDs with vulnerable population groups. Furthermore, GHA shared the Liamputtong's reading materials on how to conduct FGDs with all research team members, and she led the first two FGDs to demonstrate moderation.

## Sampling and recruitment

The qualitative approach adopted in the present study included conducting FGDs with Syrian refugee participants and in-depth interviews with key informants. The data triangulation from the FGDs and in-depth interviews provided a complimentary perspective on the findings, thus increasing the study's credibility.

For the FGDs, a purposive convenient sampling approach was used to recruit Syrian refugee participants. Refugees were purposively selected to be 18 years and older, residing in Lebanon for more than 3 months, and belonged to one of the three categories below (each further divided into groups of men and women): 1- refugees registered or recorded with UNCHR receiving formal assistance and healthcare subsidies (4 FGDs with women and 3 FGD with men), 2- refugees registered or recorded with UNHCR but not receiving any assistance for more than 6 months (3 FGDs with women, 2 FGDs and 1 structured interview with men), and 3- unregistered and unrecorded refugees not eligible for assistance or healthcare subsidies (3 FGDs with women and 4 FGD with men). These three groups are, hereon referred, to as: registered with assistance, registered without assistance, and unregistered refugees. Conveniently, our local field partners 'Action Against Hunger' (*Action Contre La Faim International*, ACF) facilitated access by assisting the research team in securing clearances to visit ITS in the Baalbek-Hermel area and providing logistical support through connecting the team with the ITS gatekeepers, also known as "shaweesh". The latter are considered prominent figures within their Syrian refugee communities and serve as gatekeepers to access the ITS and contact refugees [35]. It is worth noting that neither the "shaweesh" nor the ACF team were involved in the direct recruitment of Syrian refugees or in the data collection to avoid any undue influence or coercion.

In addition, the research team used the in-depth interviews approach to collect data from representatives working in international non-governmental organizations (iNGO), local non-governmental organizations (NGOs), and other governmental representatives operating in the Beqaa region of Lebanon (see Table 1). Snowball sampling method was used to recruit key informants for the in-depth interviews, starting with ACF staff who then connected the research team with potential local or international NGOs working with refugees in the Beqaa

**Table 1. List of non-governmental and governmental organizations that participated in the in-depth interviews in the study.**

Action Contra La Faim (ACF), Beyond organization,Counselling and Legal Assistance at the Norwegian Refugee Center (ICLA), Doctors without Borders (MSF), Gruppo Di Volontariato Civile (GVC), Lebanese Ministry of Social Affairs (MoSA), Lebanese Protection Consortium (LPC), Lebanese Red Cross (LRC), United Nations High Commissioner for Refugees (UNHCR), United Nations Development Programme (UNDP).

region. These NGOs, in turn, connected the team to additional NGOs. Key informants were contacted by email to schedule a preferred date and location for conducting the interviews. A total of 12 interviews were completed, after which saturation was reached. These interviews were conducted with multiple staff members at different levels such as field officers, program managers, technical experts, and across a variety of sectors including protection, health, food security, livelihoods, and basic assistance.

## Data collection

Data collection was conducted over 8 months. FGD were conducted during the months of November and December 2018, whereas the in-depth interviews were carried out in May and June 2019. The time-lapse between the FGD and in-depth interviews was due to severe weather conditions and time needed to obtain permission from respective NGOs to contact their officers.

For refugee participants, oral, rather than written, consent was considered due to potential security fears associated with the lack of legal status among unregistered refugees residing in Lebanon. Oral consents were obtained from all refugee participants prior to the start of each FGD and after describing clearly the study objectives, the voluntary nature of their participation, and the confidentiality of the discussions. There was no compensation for participation, except the provision of refreshments. Participants were also assured that they could refuse to continue their involvement in the study at any time without affecting their relationship with AUB, ACF, or any other agency providing humanitarian aid or services to refugees. The permission of refugees to audiotape the discussion was also secured prior to start the recording. The discussion was all conducted using general, non-professional Arabic terms to ensure clarity and ease of discussion. All transcripts and audio recordings were saved on password-protected computers with access granted only to the research team.

**Questionnaire.** Prior to each FGD, refugee participants were asked to complete a short questionnaire with the help of the research team. This method triangulation increased the confirmability and dependability of the findings, thus its rigor [36]. The moderator assisted illiterate participants in completing the questionnaire.

The questionnaire (S1 Appendix) included sociodemographic information (age, education, casual labor, household conditions), and the Arabic-translated, previously validated version of the Household Food Insecurity Access Scale (HFIAS) [37,38]. The sociodemographic questionnaire, including the HFIAS scale, were used in previous studies conducted in Lebanon with Syrian refugees and Lebanese households showing good and reliable results [26,39,40]. The HFIAS reflects the food insecurity experience of the participants and their family members in the past four weeks, and it is comprised of nine questions on the occurrence and frequency of food insecurity experience during that period. To calculate the HFIAS score per household, a standard scoring procedure was used whereby 0, 1, 2, 3 points were allocated for "non-occurrence", "rare", "sometimes", and "often", respectively. The total HFIAS score ranged between 0 and 27, with higher scores indicating higher levels of household food insecurity. Scores were used to categorize households into four levels of food insecurity (food secure, mildly, moderately or severely food insecure) depending on the number of positive responses to questions related to severe conditions, and as per the HFIAS measurement and indicator guide [25]. For the purpose of analysis in this paper, the food secure and mildly food insecure households were merged. In addition, the short questionnaire asked respondents about the frequency of their household's consumption of eight different food groups over the previous seven days to calculate their food consumption scores (FCS). The FCS is a composite score based on the dietary diversity, food frequency, and relative nutritional importance of different

food groups with higher scores referring to higher dietary diversity and frequency of consumption among households [41]. The FCS was calculated for each household and later categorized into three groups (poor, borderline and high), as per the standard thresholds [27]. Briefly, the FCS is an index that was developed by the WFP in 1996 and it has been incorporated since then into surveys to assess the food security and dietary diversity of vulnerable communities and households [42]. In Lebanon, the FCS represents a key component of the WFP annual surveys conducted to assess the food security status of Syrian refugees [3,10,43].

**Focus groups.**   Liamputtong's traditional approach [44] for conducting focus groups was adopted in the present study. For cultural appropriateness, gender congruence was observed, i.e. men moderators facilitated the men's FGDs and women moderators for women's groups. Each FGD consisted of 4 to 10 participants and was conducted in a communal room inside the ITS for a duration of one hour and a half using a FGD guide (S2 Appendix). At the start of FGDs, moderators introduced themselves and explained their roles that consisted of guiding the discussion while the observers examined the group dynamics and took notes. Concerning credibility, the moderators and interviewers conducting the FGDs and in-depth interviews shared the same first language as the participants. The decision to conclude data collection was done once we reached data saturation by group (registered with assistance, registered non-assistance, and unrecorded) and by gender within each group.

**In-depth interviews.**   Informed oral consent was secured from all key informants to ensure voluntary participation and confidentiality. The permission to audiotape the discussion was secured from participants prior to the start of the interview. Those who consented were interviewed by the moderators (D.N, H.I or F.AH) in a private setting at each of the designated key informant's offices or at the Global Health Institute (GHI) offices in Beirut, except for one interview that was conducted via SKYPE to meet the interviewee's preference. The interviews were conducted by the moderators using an interview guide (S3 Appendix). The interviewer (s) introduced themselves and re-iterated the purpose of the study in addition to emphasizing the need to understand the effect of registration status on the livelihoods of Syrian refugees. The non-participant observer recorded notes and intervened when in need to clarify certain terms or legal implications.

## Data analysis

**Qualitative analysis.**   Audio recordings were transcribed verbatim concurrently with the data collection. The research team avoided translation while transcribing to observe fidelity of meanings [45]. Instead, data were analyzed in Arabic, and only the quotes that were used in the manuscript were translated from colloquial Arabic to English. After the completion of data collection, an inductive thematic analysis using Braun and Clarke approach was used for both interviews and FGD [46]. After the analysis was completed, all quotes were translated into English.

Each FGD participant was given an ID during coding. The first letters described the gender of the participants, with 'M' denoting men and 'W' for women. The second letter referred to the registration status, where 'R' was for registered and 'NR' for the unregistered. For registered participants, 'A' was for those receiving assistance and 'NA' for those not receiving assistance. The first number was the number of the focus group, and the second was for the number of the participant preceded by a P (e.g. P2). For FGDs, and when we were unable to track each participant's statements, the participant identifier was excluded. For example, M.R. NA.1 was a man, registered, not receiving assistance from the FGD, with the exact participant unidentified. As for codes from the interviews, the source of the quote was indicated by the number of the interview and the affiliation of the key informant, for example, iNGO referring to an interview with a representative from an international NGO.

Data was thematically analyzed along six phases. In phase 1, two members of the research team read and re-read each transcript to get acquainted with the information. In phase 2, descriptive coding line-by-line for each transcript was conducted. In phase 3, an initial list of all codes was generated by the two team members (D.N. and A.H.). In phase 4, the research team discussed the relationships between codes. A log of potential themes and sub-themes was developed, including a list of definitions and quotes to illustrate each theme and sub-theme. In phase 5, the list of candidate themes and sub-themes was further refined based on consensus among all research team members and to highlight the existing relationships between them. As for phase 6, the findings were presented in a narrative form, and a synthesis of the results was included. These findings were supported with quotes from interviewees and beneficiaries relating to identified themes and sub-themes.

**Quantitative analysis.**   Data from the short questionnaire completed with participants at the beginning of the focus groups were entered and analyzed using Statistical Package for the Social Sciences (SPSS) version 22 software [47]. Analysis of variance (ANOVA) with Tukey's post-hoc analysis and chi-square analysis were conducted for continuous and categorical variables, respectively. Results were reported as means and standard deviations (SD) for continuous variables and as frequencies and percentages [n (%)] for categorical variables. Statistical significance was reported at p-value $\leq$5%.

**Integrated analysis.**   Analysis of integrated findings between FGD and short questionnaire were integrated by contiguous narrative approach where the quantitative findings were presented first followed by the qualitative findings. Findings were further discussed for concordance, discordance or expansion. Integration of results was assessed by the research team [48]. Findings were reported based on Good Reporting of A Mixed Methods Study (GRAMMS) criteria (S4 Appendix) [49]. Data was analyzed manually.

## Results

### Sociodemographic characteristics

A total of 122 Syrian refugees participated in 19 FGDs within the present study, and there was a total of 40 men and 82 women participants. Descriptive characteristics of the study sample, including socio-demographic data and household food security status, were presented in Table 2.

The Syrian refugee participants in our study were residing in Lebanon for an average of 4.6 years. The majority of refugee participants had a primary school education or below (63%) and did not engage in casual labor (75%). Most participants were married (88%) with an average of 5 household members and a mean crowding index of 4 persons/room. Moreover, more than half of Syrian refugee participants and their families were found to be severely food insecure (65%) and had poor or borderline food consumption (55%).

Significant differences were observed by registration and assistance status for several variables, including years of stay in Lebanon, number of children, crowding index, household food insecurity and food consumption. Registered Syrian refugees with assistance were found to have a significantly higher crowding and total number of children compared to other groups (p = 0.002 and p = 0.001). With regards to household food security status, the majority of the unregistered Syrian refugees and those registered without assistance were found to be severely food insecure compared to refugees receiving assistance (83.3 vs 78.6 vs 43.1%, respectively), and the difference was statistically significant, p<0.001. In addition, a higher proportion of unregistered Syrian refugee households and those registered without assistance were found to have poor or borderline food consumption scores as compared to registered households receiving assistance (p<0.001).

**Table 2. Descriptive characteristics of Syrian refugees in the total sample and by registration and assistance status.**

| | Total (n = 122) | Registered with assistance (n = 52) | Registered without assistance (n = 28) | Unregistered without assistance (N = 42) |
|---|---|---|---|---|
| **Years in Lebanon**[**] | 4.61±2.34 | 5.20±1.42[a] | 6.67±2.06[b] | 3.03±2.38[c] |
| **Educational Level** | | | | |
| Primary school or less | 77(63.1) | 33(63.5) | 16(57.1) | 28(66.7) |
| Middle School or higher | 45(36.9) | 19(36.5) | 12(42.9) | 14(33.3) |
| **Marital Status** | | | | |
| Unmarried/divorced/widowed | 15(12.3) | 7(13.5) | 5(17.9) | 3(7.1) |
| Married | 107(87.7) | 45(86.5) | 23(82.1) | 39(92.9) |
| **Casual Labor** | | | | |
| Yes | 31(25.4) | 15(28.8) | 3(10.7) | 13(31.0) |
| No | 91(74.6) | 37(71.2) | 25(89.3) | 29(69.0) |
| **Number of household members**[*] | 5.00±2.66 | 5.71±2.11 | 4.43±2.79 | 4.54±3.02 |
| **Number of children**[*] | 2.58±2.47 | 3.48±2.16[a] | 1.93±1.63[b] | 1.90±2.95[c] |
| **Household crowding index**[§][**] | 3.60±2.08 | 4.37±1.99[a] | 3.10±2.27[b] | 3.14±1.8[c] |
| **Household food security status**[‖][**] | | | | |
| Food secure/ mildly food insecure | 4(3.3) | 3(5.9) | 1(3.6) | 0(0.00) |
| Moderately food insecure | 38(31.4) | 26(51.0) | 5(17.9) | 7(16.7) |
| Severely food insecure | 79(65.3) | 22(43.1) | 22(78.6) | 35(83.3) |
| **Household food consumption score**[ᵩ][**] | | | | |
| Poor (0–21) | 24(19.7) | 4(7.7) | 8(28.6) | 12(28.6) |
| Borderline (21.5–35) | 43(35.2) | 8(15.4) | 11(39.3) | 24(57.1) |
| Acceptable (>35) | 55(45.1) | 40(76.9) | 9(32.1) | 6(14.3) |

†Continuous variables are presented as means and standard deviations (SD) whereas categorical variables are presented as frequencies and proportions (%).

* Significance at p≤0.05

** significance at p≤0.001.

a, b, c Different superscripts were significantly different among groups using ANOVA Tukey post hoc analysis.

§ Crowding index was calculated as the total number of household members divided by the total number of rooms in a household (excluding kitchens, bathrooms and balconies) [50].

‖ Household food security status was assessed using the Household Food Insecurity Access Scale (HFIAS) measurement and indicator guide [37].

* Fisher's exact test was used for cells with count less than 5.

ᵩ The Food consumption Score (FCS) was calculated using the frequency of consumption of different food groups consumed by a household during the prior 7 days. Using standard food groups and standard weights, the frequency weighted diet diversity score was computed and then categorized into three groups based on standard thresholds [41].

## Emergent themes

Two main themes with eight sub-themes emerged from the FGDs and key informant interviews (Fig 1). The first theme, namely formal and informal livelihood strategies adopted by refugees, included four sub-themes: 1- dependence on cash assistance for survival; 2 informal employment and exploitative work conditions; 3- child labor & early marriage; and 4- accruing debt & social exchange. The second theme referred to the social determinants of health and limited access to health services with three common subthemes, namely: 1- high food insecurity and low diet diversity; 2- overcrowding & poor living conditions; and 3- limited access to healthcare. Overall, it was noted that although these sub-themes were common among the three groups of refugee participants (registered with assistance, registered without assistance, and unregistered refugees), the severity of these coping strategies and their adverse consequences were particularly evident amongst the most vulnerable unregistered group.

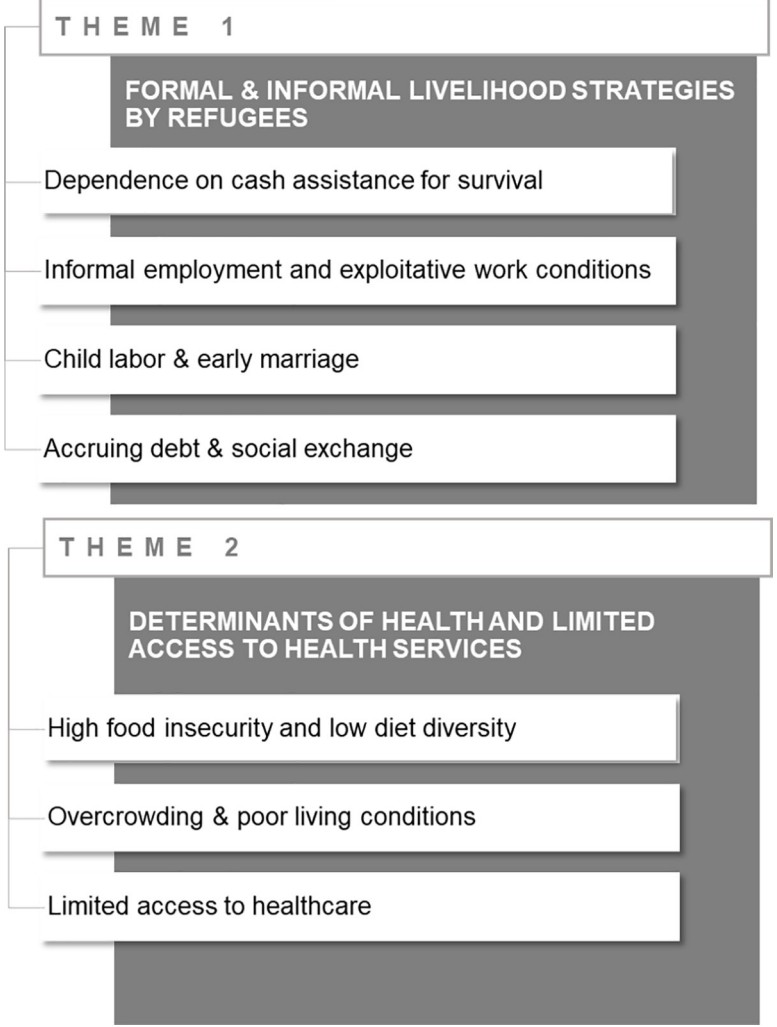

**Fig 1. Themes and subthemes emerging from the focus groups and key informant interviews.**

## Theme 1: Formal & informal livelihood strategies

**Dependence on cash assistance for survival.** All registered refugees who were receiving assistance reported food and/or MPC assistance as their main and essential sources of livelihoods, "*God help those not receiving assistance*" (W.R.A.2: P: 10), despite being insufficient and unsustainable. Registered refugees not receiving assistance and unregistered refugees also highlighted the importance of assistance, and believed their situations would improve if they were to receive any type of assistance by UNHCR. *"If I can get anything, if I can get only $260 (referring to the cash assistance), I will be able to get by"*; *"The registered have everything secured for them, every month they receive fuel, packages, cash, everything."* (M.NR.4). Key informants confirmed refugees' dependence on assistance for their livelihoods and survival. However, the majority of these informants clarified that the assessments of refugees' needs, and accordingly assistance allocation, under most projects were not based just on registration status, but also on their level of vulnerability. Key informants also echoed the fears of refugees that the current programs providing assistance to refugees were short-term, and that long-term, sustainable projects are not feasible in the Lebanese context due to donor or governmental restrictions:

"*You cannot do anything long-term because long-term means permanent, permanent means that [refugees] are going to stay here and we go into a political conflict*" (iNGO—Int. 3).

**Informal employment and exploitative work conditions.**　Employment was regularly identified as an important source of livelihood among all the participants, especially among those not receiving assistance (whether registered without assistance or unregistered). In fact, all of the participants were informally employed (i.e. without work permits), mainly in construction, agriculture, and tobacco picking, with an average daily salary of $4. There were a couple of cases of women cleaning houses and one case of garbage collector, but these were only reported amongst unregistered refugees. Regardless of registration status, work opportunities were highly irregular, often provided and controlled by the gatekeeper, known as "shaweesh", and seasonal, mostly restricted to the agricultural season in the summer. These findings were consistent with those reported by the NGO representatives, who echoed the engagement of refugees, regardless of their registration status, in informal employment without legal work permits as their source of livelihoods.

Furthermore, FGD participants reported numerous cases of work exploitation, mostly pertaining to lack of payment by their employers. One participant said: "*The employer tells you to go away and I'll come pay you later, but he doesn't come; I mean here we are deprived of our rights and there's nothing we can do, we have no support or back-up*" (M.R.NA.1). Key informants echoed the poor and exploitative work conditions of refugees, whether through being underpaid by their employers, controlled by the gatekeepers "shaweesh", or abused verbally and physically: "*The employer rarely deals with these people. It's usually the Shaweesh that beats them with a stick sometimes, if he doesn't beat them, there's the verbal abuse that they suffer all day long, and the emotional effect.*" (iNGO—Int. 3). Some key informants reported that Syrian women resort to socially less acceptable jobs, such as sex work and begging: "*So it is more of a coping mechanism for women who are willingly doing that, but for younger girls it is sometimes a coping mechanism and sometimes exploitation against their will*" (Local NGO—Int. 9). Nevertheless, sex work and begging were not reported by refugees during the FGD.

**Child labor and early marriage.**　Children as young as 11 years old were essential to the livelihoods of their families, regardless of registration status. "*The children help, they help the mom in work*" (W.R.NA.2); "*12 year olds can work, but if they are 7 years old or younger they can't work in anything*" (W.NR.2). According to key informants, child labor is widespread and normalized in Lebanon, further increasing rates of children who were out of school. One of the main reasons mentioned by key informants was that children under 15 years of age are not required to have legal papers in Lebanon, and as such they can easily cross checkpoints and are better positioned to be working than their parents." One key informant also mentioned the preference of employers in hiring children in certain cases: "*As of 6 years old they send them to work, especially I don't know if you know about Marijuana but you have to weed it by hand, there are no pesticides, they prefer small size people, because they don't damage the crop and they are near the ground to do it with their bare hands or with a small shovel. . ., imagine the smell they are smelling all day, I wonder if they get dizzy or sick*" (iNGO—Int.3).

Regardless of registration status, early marriage was another subtheme that emerged from FGD, and it was considered a culturally acceptable practice among many Syrian refugees. According to the key informants, early marriage was a strategy utilized to alleviate the financial burdens faced by families, as marrying girls earlier decreases family expenses. Moreover, few unregistered men in the FGD expressed their need for a young girl to help them around the household: "*There was no one to help my mother, so I got married I had to get married. My mother is an older woman, she has high blood pressure, diabetes and poor eyesight, you know what I mean. . . I want someone to work instead of my mom, I have 2–3 younger sisters, yes I want a girl to help my mother, I married her and now she helps her*" (M.NR.4).

**Accruing debt and social exchange.** All FGD participants, regardless of whether they received assistance or not, agreed that accruing debt was another source to support their livelihoods. "*I swear we are all in debt, there isn't one household who is not in debt*" (M.R.A.1) said one registered man who was receiving assistance. Syrian refugees reported being in debt to shopkeepers, pharmacists, neighbors, property owners or "shaweesh" (ITS gatekeepers), and the level of debt of refugees accumulated more in winter due to lack of seasonal employment. In addition, the magnitude of debt was significantly higher amongst unregistered refugees: "*We request debts to survive, up to 1000$, what can we do, we need to live regardless of excessive debt*" (M.NR.1). Registered refugees with assistance who were working were able to repay some of their debts, whereas it was very challenging for their unregistered counterparts to pay off their dues, thus accumulating larger amounts of debts.

Dependence on debt was also echoed by key informants as a negative livelihood strategy and coping mechanism, particularly amongst refugees who were unregistered: "*It is true that debt is common regardless of registration status, but for the others (referring to those receiving assistance) every month they pay back some of their debt, the problem is for those unregistered with the UNHCR or registered and not benefitting from assistance, their debt is very big, large, very, and this affects a lot of things such as sending their children to work, sending the wives for extra work, the type of work becomes more difficult because the number of unregistered refugees is increasing, and as such work opportunities are decreasing so they go to work on the roads in prostitution for example and other jobs. The debt is a big problem because all Syrians are now in debt*" (Local NGO—Int. 9).

In addition to accumulation of debt, unregistered refugees expressed reliance on more positive mechanisms such as social exchange within Syrian refugee families and among friends and neighbors to share food and/or assistance. For example, few unregistered refugees and their registered parents, or elderly unregistered parents and their registered children reported often sharing assistance: "*I have children, I let my mother and father bring me food assistance cards and diapers for the children*" (M.NR.4). However, some unregistered participants living away from their families believed that only other unregistered neighbors were willing to share their food and help each other because they understood each other's difficult situations "*Today my situation is under zero for example? I have 2 cups of sugar, Fatima (unregistered neighbor) comes to borrow a cup of sugar, I will keep a cup and give her the other*" (W.NR.2). Registered refugee participants rarely mentioned sharing of assistance between community members and only a few of the international and local NGO representatives referred to the social exchange of assistance between refugees.

## Theme 2: Determinants of health and limited access to health services

**High food insecurity and low dietary diversity.** All refugee participants, regardless of assistance and registration, reported difficulties in securing food for the family and poor dietary diversity. Occasionally, refugee women or children working in agriculture were able to take home leftover vegetables. To overcome food shortages, refugees resorted to different coping strategies such as buying cheaper, but low-quality, foods and over time decreasing even the quantity of food consumed. Food diversity was worse among refugees without assistance (registered or unregistered), whereby the latter would frequently skip meals and were more likely to suffer severe food insecurity or even occasionally eat expired or rotten foods. "*There are times where we have no lunch and no dinner, there are other times where we only eat bread and tea*" (M.NR.1). At times of prolonged food shortages, adults would skip meals to prioritize their children needs. *"For example if the kids don't feel full, we say let the kids eat until they're full, we can handle hunger but the kids can't so we give them the food, they are the priority"* (M.NR.4).

Interviews with key informants confirmed these findings. When discussing skipping meals as a coping mechanism, one local NGO representative said, *"It is standard now eating 2 meals a day. It's become a daily reality [for refugees], their behavior has changed and it's not a temporary coping mechanism anymore."* (Local NGO—Int. 6).

**Overcrowding and poor living conditions.**   Overcrowding was a common concern among participants regardless of registration status. "*We are 7 people in this tent, in this room, in this 1 room, we all sleep in it*" (W.R.A.2). In addition, participants reported overall poor and unsanitary living conditions and that the availability of proper waste disposal systems were inconsistent between ITS. In order to cope with these challenges, many participants had to dig unsanitary holes in the ground that they would discharge manually. Participants also reported a high incidence of gastrointestinal infections amongst children attributed also to poor water quality. "*It is possible that the landlord will evict us due to the holes we dig as bathroom replacements, because we cannot dig a new hole every other day, I mean we suffer from this, it is our biggest problem, it causes sickness*" (M.R.A.2). Refugees also experienced extremes in temperatures due to poor housing, as their shelters were very hot in the summer, and cold in winter. The cold weather was commonly dealt with by burning fuel, plastic or any available item such as clothes, resulting in respiratory complications amongst children.

Key informants confirmed many of the experiences reported by the refugees in terms of the poor and unsanitary living conditions that were causing health issues and breakouts "*We stopped our desludging for like a month and the situation is deteriorating in some sites; health breakouts like scabies and other health issues at the community level. Management of grey water is not a priority although we are now witnessing some sites that are highly under risk because of that and it's causing like a plague of rodents, rats, and no activities are being done for this regard.*" (iNGO—Int.5).

Another challenge that a few refugee participants expressed is their constant fear of eviction from their tents due to their inability to cover rent for the land property owners: "*You don't know what day you will get evicted, in the extreme cold, in winter he (referring to the landlord) would evict us*" (W.R.NA.1). A few unregistered refugees lived on their employers' land or warehouses where they had to work anytime in return for housing and faced eviction threats if they did not work upon their employer's request. One of the registered women, who was not receiving assistance and was living in an employer's warehouse, explained: "*It's mandatory, my husband used to work for him, you know what I mean? I mean instead of payment he gives us shelter, but now there is no work, he says I want to evict you*" (W.R.NA.1). As a coping mechanism, both registered and unregistered participants shared their tents with relatives and friends when they first arrive to Lebanon. It is worth noting that the sharing of housing was often a transition or temporary situation amongst registered refugees. "*Of course, for example if someone from your family comes from Syria where is he going to stay? He needs to wait until the UNHCR gives him assistance so of course he is going to live with his family in the same tent until the UNHCR gives him assistance*" (M.R.NA.1). On the other hand, unregistered refugees reported the sharing of housing with other refugees as a more permanent living arrangement: "*If I go back to my tent I would die of cold, that's why I am living at my parents*" (M.NR.3: P5).

**Limited access to healthcare.**   There was a consensus among refugees that registration with UNHCR determined access to healthcare: "*those who have a code [registered] get helped with the payment, they are much better off*" (M.NR.3). However, out-of-pocket payments for hospital admissions and transportation fees were among the main barriers to access health care in Lebanon among all refugee participants. Refugees reported that they were required to pay upfront to be provided care at hospitals: "*the hospital needs $3000 only as an insurance, but she is not allowed to enter; even if you tell them she will die, they still say she can't enter!!*" (M.R.A.3). Although healthcare subsidies were provided for registered refuges, out-of-pocket costs

for antenatal care and deliveries were barriers to all refugees: "*It's expensive, especially for Syrians, you always have to pay out of pocket*" (W.NA.R.3). One registered woman narrated her experience resorting to a home delivery due to the high out-of-pocket payment for hospital delivery.

Many registered and unregistered participants stated that discrimination and poor treatment in hospitals by healthcare professionals were also a common experience: "*When I went to get the operation, they were yelling at me, and when I was done, they lifted me and then threw me on the bed, and even under anesthesia you feel the pain, you feel the pain of being thrown on the bed*" (W.R.A.1. P5).

Due to financial constraints, registered and unregistered refugee women reported prioritization of medical conditions. For example, medical issues that were not perceived as crucial and urgent, such as reproductive tract infections, were almost always ignored without resorting to medical attention and instead were treated with home remedies "*I swear the most important issue we have been facing is reproductive tract infections, and I can only rely on God because the cheapest medication for infection costs $20, I swear it's very expensive so we just stay sick*" (W.R.NA.2. P.6). This was echoed in interviews with key informants: "*Not going to the necessary checkups, not taking medications all together because they cannot afford it. What is really sad is that sometimes people are sick, but they don't know because they don't get checked up.*" (Local NGO—Int. 6).

Another coping mechanism for overcoming financial barriers to health services was seeking alternative informal sources of healthcare, such as pharmacists for medical advice, as it was much cheaper and easily accessible. "*There's no one other than pharmacies*" (W.R.NA.P9). Many participants also opted to acquire their medications from Syria where prices are much lower, either done firsthand or through an intermediary. Some participants mentioned that individuals occasionally went to Syria to get medical care and then came back into Lebanon illegally. "*We asked for assistance to cover the surgery, but we don't have papers, we couldn't do it, he had to go to Syria, and he has to serve in the army, so they took him and he had to stay*" (W.NR.3.P4). These alternative sources of healthcare were also reported by local and international NGOs: "*Some Syrians sometimes prefer to go to Syria to put their selves at risk to do some surgeries and come back. For example, there are some women who are going for C-sections and they are putting themselves at risk because some of them are undocumented*" (iNGO—Int. 8).

A few participants mentioned switching IDs between registered and unregistered participants in order to access medical services, "*For example my son today is not registered, and it is necessary to get him to the hospital, I'm going to have to admit him under my nephew's name, he's the same age and registered. It's a rare case but these are our circumstances, it's better than having my son die, I'm going to have to do this, this is not acceptable for the hospital nor for us but we are forced to do this what can we do*" (M.NR.2). Only one key informant mentioned this coping mechanism: "*We deal with cases of registration you sometimes you will hear about cases where women using the ID of her sister to give birth in the hospita*l" (iNGO—Int. 7).

**Integrated analysis.**   Integration of qualitative and quantitative data, albeit being modest, revealed mostly concordance specifically on food insecurity challenges. According to the questionnaire results, the majority of Syrian refugees were experiencing moderate to severe food insecurity and had poor or borderline food consumption, with the highest percentages observed among those who were unregistered followed by those who were registered without assistance. Similarly, FGD findings echoed the challenges faced by Syrian refugees in securing food for their families due to the high food prices in the market, particularly among those who were unregistered/unrecorded.

For employment, discordance was noted between the FGD and questionnaire findings. While FGDs and interview results reported almost universal informal employment, results

from the questionnaire showed that only 25% were involved in labor. This discrepancy can be interpreted either due to the seasonality and irregularity of work opportunities, especially that the data was collected in the winter, which is considered a low agricultural season, or perhaps due to the fear of refugees from disclosing this information in the questionnaire as their employment was informal and illegal (i.e. without work permit).

## Discussion

The present study explored the livelihood strategies and coping mechanisms of Syrian refugees in the Beqaa region in Lebanon and their impact on health, taking into consideration refugees' registration status and access to assistance. To our knowledge, this study is the first to unravel the livelihood strategies of unregistered Syrian refugees, a hard-to-reach group that is arguably among the most vulnerable, yet least explored segments, of the displaced population.

Results showed that refugees were highly dependent on formal assistance for survival when received, albeit being insufficient. Also, regardless of registration status, refugees resorted to informal livelihood strategies to improve their living conditions including informal employment, child labor, and in some instances, early marriage for girls. These findings were consistent with previous studies conducted to assess the challenges faced by Syrian refugees in Lebanon [10,51], yet the present study is the first to highlight the severity and extent of these mechanisms, particularly among unregistered refugees. According to the 2019 Vulnerability Assessment for Syrian refugees (VASyR) survey conducted to assess the needs of registered refugees in Lebanon, 97% of refugee households resorted to some type of livelihood coping strategy to make up for their economic hardships including selling productive assets (10%) and reducing expenditures on health (54%) and education (30%). In addition, the same survey showed that some refugee households resorted to more severe strategies, such as withdrawing children from school (12%), engaging school-aged children in income generation (5%), and marriage of children under 18 years of age (1%), all of which may affect the long-term coping capacity and the overall wellbeing of refugees [10]. Another study conducted in the Beqaa valley in Lebanon showed that more than half of the Syrian refugee children (4–18 years of age) from 1902 refugee households were engaged in some form of child labor, predominantly in the agriculture sector, and almost 20% of these working children had illnesses or disabilities [52]. Child labor for income-generation is a phenomenon witnessed in various refugee contexts, and it raises serious concerns with respect to heightened risk of exploitation and abuse that children may be subjected to, and its implications on their health and wellbeing on the short and long-term [3,52,53].

In addition, early marriage was reported by Syrian refugees and key informants as a mechanism to alleviate the economic hardships of their families; however, only a few key informants reported that refugees resorted to sex work and begging in certain dire situations. Early marriage has been previously argued as a culturally acceptable practice among Syrians even prior to the war, a phenomenon that was further adopted by Syrian families post-displacement to overcome economic and security challenges and to give adolescent girls an improved and more secure life [54–56]. According to Bartels et al (2018), in many instances, Syrian refugee girls were ready to accept the proposals from well-settled men to secure more protection and escape the hardship conditions of their families [57]. The dire living conditions, poverty and insecurity were also reported in other studies as key motives for the households to take that decision on the behalf of their daughters and marry them off at an early age [51,54]. Sex work is another informal coping mechanism that was adopted by Syrian refugees to overcome their economic hardships, yet it was reported only by the key informants in the present study. Given the sensitive nature of this practice and the associated cultural, religious and legal taboos

of sex work [3,52,53], it was not surprising to observe a discrepancy between Syrian refugees and key informants on the reporting of sex work as a form of informal employment. Nevertheless, these findings highlight the need for further exploration of sex work as a form of informal employment among this vulnerable population.

Accumulating debts was another common negative livelihood strategy reported by our study participants adding to the refugees' vulnerability and compromising their health. Refugees not receiving assistance reported that they continued to accumulate debt, whereas their refugee counterparts who were registered with UNHCR and receiving assistance were able to cover some of their debt at the end of each month. In line with our study findings, 9 out of 10 registered Syrian refugee households in the VASyR were in debt, and the main reasons for borrowing money were for food, rent and health care payments, which shows that even registered refugees continue to lack resources to cover their essential needs [10]. While debt is a formal strategy meant to maintain normalcy and dignity by increasing the purchasing power for essential items, such as nutritious food, i.e. fruits and vegetables, or securing a shelter; it compromises the psychological status of individuals by adding another layer of complexity and stress to the fragile status of refugees [58]. Social exchange of assistance, goods and services, on the other hand, was noted as an acceptable livelihood strategy among the Syrian refugee community within their respective ITS. For example, refugees who were registered with UNHCR and receiving assistance were found to share their food and other resources with other family members, who were unregistered and with difficult financial situations. The use of social exchange has been previously reported as a much needed form of assistance at times of difficulty whether at the financial, personal and health levels that can increase the resilience of the communities [59,60].

Another major theme that emerged from the FGDs and in-depth interviews were the poor determinants of health and limited access to health care services experienced among refugees, particularly those who lack formal assistance. Food insecurity and limited dietary diversity, as well as overcrowded and unsanitary living conditions were common challenges among all refugees in our study sample. These conditions can have serious repercussions on the health status of refugees, as reported by the participants, and have been previously associated with increased morbidity and mortality particularly among vulnerable groups, such as young children, pregnant and lactating women, as well as older adults and those with disabilities [61–63]. The negative health consequences of these social determinants of health are further aggravated by limited access to healthcare [64]. The healthcare system in Lebanon is fragmented and privatized, with multiple studies reporting on the common barriers for access among refugees, which include out-of-pocket costs, transportation, distance, poor treatment and discrimination and security concerns [64]. It is true that refugees registered or recorded with UNHCR benefit from subsidies on certain health services, such as primary healthcare, antenatal care and deliveries, and emergency hospital fees. However, findings from the present study highlight that out-of-pocket costs remain a barrier even for registered refugees. Most refugees would spend the little they have on basic necessities such as food and shelter [63,65]. Healthcare becomes a luxury and individuals resort to prioritization of medical conditions for which to seek healthcare. As a result, refugees seek alternative sources of healthcare such as home-remedies, informal healthcare providers [66], and pharmacists [63], as evidenced previously by 35% of registered Syrian refugees in the Baalbek-Hermel governorate seeking primary care from pharmacies [10]. Refugees in the present study also reported increasingly acquiring their medications from Syria [63] and occasionally travelling to Syria for surgical procedures despite associated ramifications. It is noteworthy to indicate that two decades prior to the crisis, the health care system in Syria made several strides in terms of improving access to health care, reorganizing health services, enhancing data metrics, and starting plans to address chronic

diseases [67]. Although these achievements were destroyed at the onset of the Syrian war, and access to healthcare became severely restricted together with deterioration in the quality of care, Syrian refugees in Lebanon were still willing to travel back home to seek more affordable health services [68].

## Implications for humanitarian systems and potential recommendations

Our study findings shed light on the overall vulnerability of Syrian refugees in Lebanon and the economic and health disparities among refugees based on registration and assistance provision. Despite large-scale humanitarian assistance and extensive efforts exerted by GoL with UN agencies, as well as local and international partners and donors, the economic situation of the Syrian refugees in Lebanon remains precarious. Refugees continue to resort to a multitude of formal and informal livelihood strategies and negative coping mechanisms to provide for their basic necessities. Refugees, in Lebanon as with other humanitarian settings, are categorized based on legal status, registration and vulnerability, with each label carrying a significant weight. Vulnerability entails much meaning in the lives of refugees, as it is the main stratification category used by UNHCR and many other organizations to prioritize assistance allocation [69,70]. The level of vulnerability is determined based on a number of set characteristics, which include household demographics, dependency, shelter conditions, and income [69,70]. However, a question can arise as to what exactly characterizes a vulnerable refugee in Lebanon and other protracted conflicts when most displaced individuals are living below the poverty line and are utilizing negative coping mechanisms of varying levels. Others argue that vulnerability as a criterion for assistance allocation and interventions is problematic due to its inherent exclusion of individuals from assistance [70]. In Lebanon, the sample studied in major assessments conducted by humanitarian agencies, including VASyR, are primarily composed of registered and recorded refugees [69]. Arguably, the most vulnerable population of unregistered refugees is left out of these assessments and subsequently humanitarian aid [69]. In addition, and as with other countries hosting Syrians during an ongoing war, refugees may chose not to register with UNHCR out of fear of disclosing their identities and place of residence to authorities [65]. Other reasons can be logistical, including the lack of proper documentation and legal paper work in the country, cost, transportation, and other barriers [71]. Nevertheless, these reasons should not prohibit the humanitarian, public health, and research community from assessing and addressing the needs of all marginalized and vulnerable populations.

Our study findings highlight that assistance mattered but remains insufficient to meet the basic needs of Syrian refugees in the country. The inability to meet basic needs by refugees has a compounding effect on health, as food insecurity and poor living conditions exacerbate the health of refugees, and healthcare access becomes deprioritized. These findings raise the question as to whether the current approach for assistance is effective, or if there is a need for a paradigm shift in the way we consider humanitarian aid. During an acute crisis, it is expected to offer assistance for essential services, such as access to food and shelter. However, the Syrian conflict and resulting refugee crisis can no longer be treated as acute, and require alternative measures contextualized for survival and more dignified livelihoods in protracted crises. In a protracted crisis, the life and/or livelihood of a great number of individuals is in danger for an extended period of time, with official institutions unable to secure their needs [72]. This brings forward the need for a paradigm shift in addressing the needs of refugees in protracted crises [73]. A paradigm shift that takes into consideration several factors, including the protection of individuals and their human security, placing it at the forefront of all humanitarian programs and interventions rather than focusing on traditional state-centric ideas of security and on acute crisis management approaches [74]. In Lebanon, Syrian refugees continue to face

substantial challenges with respect to their registration with UNHCR (given the GoL restrictive measures since 2015), their legal status in the country and their limited integration into the workforce, which affects their employment and overall livelihoods. Thus, relevant policies and studies that can address issues relevant to the protection of Syrian refugees and their potential integration into the workforce while taking into consideration the growing complexities and challenges faced in host countries, such as Lebanon and similar LMICs, are much needed.

Another major consideration is to focus on interventions with not only efficient and effective outcomes, but also of sustainable nature that can last beyond the short-term and rapid solutions. To address food security, it is not sufficient to fine-tune the targeting of existing food assistance modalities. There is a need to examine what complimentary interventions to food and cash transfer programs are needed to improve food security of refugees and contribute to human development and wellbeing [73]. For example, local and international organizations can support community-led interventions, such as community kitchens [26] and help link farmers to schools (school feeding programs), to promote food and nutrition security while providing sustainable livelihood opportunities for the most vulnerable. These community-based initiatives also provide employment opportunities for refugees to improve their livelihoods, and to transition from informal and illegal livelihood strategies to more formal ones. Cash assistance has been shown to improve the conditions of the most vulnerable refugees in multiple humanitarian settings, including Lebanon [75]. However, in Lebanon, where cash assistance is used to cover daily basic necessities, and with lack of employment opportunities, their positive effect may not be sustained long-term [75]. Providing employment, job-skills training and investment opportunities, along with cash assistance can help alleviate many barriers experienced by refugees in securing their livelihood and provide more sustainable long-term solutions. Finally, with respect to improving access to health care, temporary policies, similarly to those adopted in Turkey [76], allowing Syrian healthcare providers to practice their professions among their refugee communities can be examined as potential approaches to relieve the high burden on the strained health care system of host communities [66]. Moreover, Syrian healthcare workers have good rapport and reputation within their own communities, thus they may be able to alleviate the financial strains and fears of social discrimination experienced among refugees when interacting with health workers from the host community.

## Strengths and limitations

The present study was the first to explore and compare the perceptions and experiences of registered versus unregistered Syrian refugees in Lebanon with regards to their livelihood strategies and adopted coping mechanisms of survival. In addition, the triangulation of data was conducted through adopting quantitative and qualitative research methods with the refugees and key informants to increase the credibility of the study findings. Nevertheless, the study has few limitations worth considering. Given that the recruitment of refugee participants was conducted using a non-random sampling approach, selection bias cannot be ruled out. Nevertheless, the purposive convenient sampling used in the present was the best possible approach to reach the various groups of refugees, including unregistered refugees. The latter group represents a hard-to-reach population living in remote ITS and are not easy to identify or locate due to security and safety concerns. Low response rate on the short questionnaire was also noted among the men and unregistered refugees. This could be explained by time constraints and the fear of disclosing sensitive information that may risk respondents to be exposed to the authorities. To overcome these challenges, study participants were assured of the confidentiality and privacy of the data. In addition, the research team members received extensive

trainings at the beginning and throughout the data collection period given the sensitive nature of the discussions and the legality concerns of the interviewed participants.

## Conclusion

In conclusion, the study findings highlight the high dependency of Syrian refugees in the Beqaa region on formal assistance, when received based on their registration status, albeit being insufficient. Refugees reported ongoing financial constraints and challenges leading to the adoption of informal livelihood strategies and negative coping mechanisms with dire consequences on their health and wellbeing. Lack of registration and formal assistance increased the vulnerability of Syrian refugees in Lebanon to food insecurity, poor dietary diversity, inadequate and unsanitary living conditions, all of which were factors contributing to poor health status. Adverse health impacts were further exacerbated by poor access to healthcare due to high out-of-pocket costs and the need to prioritize necessities such as food and shelter. Consequently, refugees resorted to alternative methods to seek care through alternative sources of healthcare and home remedies, as well as prioritizing life-threatening and serious health conditions to seek medical care. Our findings shed light on the economic and health disparities among marginalized Syrian refugees and highlight the need for more tailored humanitarian programs and services that can target the most vulnerable and hard-to-reach groups of refugees, including the unregistered and unrecorded refugees. These findings also call for integrated and sustainable humanitarian programs that address the basic food and healthcare needs of refugees while strengthening the resilience of healthcare systems in host countries. Given the protracted nature of the Syrian refugee crisis, a paradigm shift is also needed to address the social determinants of health among refugees, including the protection and legal status of refugees together with policies that can consider the temporary integration of refugees into the workforce. These policies can facilitate the access of refugees to safe and dignified sources of livelihoods, which in turn can reduce their economic vulnerability and help improve their food security levels and health status. In addition, future research is needed to examine potential interventions and strategies that aim at reaching the hard-to-reach groups and those most marginalized groups. With the ever-growing complexities and challenges faced by the forcibly displaced populations and their host countries, humanitarians, policy makers and researchers will need to work collectively to design and evaluate sustainable solutions that can promote resilience among these vulnerable communities.

## Supporting information

**S1 Appendix. Short questionnaire with Syrian refugees.**
(DOCX)

**S2 Appendix. Focus group discussion guide with Syrian refugees.**
(DOCX)

**S3 Appendix. Interview script with key informants.**
(DOCX)

**S4 Appendix. Good Reporting of A Mixed Methods Study (GRAMMS) criteria.**
(DOCX)

## Acknowledgments

Authors would like to first express their sincere gratitude to all participants and key informants who participated in this study. We would like to greatly acknowledge Action Against Hunger

for facilitating the access of the research team to the Syrian refugee beneficiaries residing in the informal tented settlements and to key informants from various local and international humanitarian organizations. A special thank you to Ms. Nour El Arnaout who assisted in developing the research tools and Mr. Abdulghani Abou Koura who assisted in field work and transcription of the focus group discussions.

## Author Contributions

**Conceptualization:** Gladys Honein-AbouHaidar, Lamis Jomaa.

**Data curation:** Dana Nabulsi, Fida Abou Hassan, Gladys Honein-AbouHaidar, Lamis Jomaa.

**Formal analysis:** Dana Nabulsi, Hussein Ismail, Fida Abou Hassan, Gladys Honein-AbouHaidar, Lamis Jomaa.

**Investigation:** Dana Nabulsi, Hussein Ismail, Fida Abou Hassan, Lea Sacca, Gladys Honein-AbouHaidar, Lamis Jomaa.

**Methodology:** Dana Nabulsi, Hussein Ismail, Fida Abou Hassan, Lea Sacca, Gladys Honein-AbouHaidar, Lamis Jomaa.

**Project administration:** Fida Abou Hassan, Gladys Honein-AbouHaidar, Lamis Jomaa.

**Supervision:** Gladys Honein-AbouHaidar, Lamis Jomaa.

**Validation:** Fida Abou Hassan, Gladys Honein-AbouHaidar, Lamis Jomaa.

**Visualization:** Dana Nabulsi, Hussein Ismail, Fida Abou Hassan, Gladys Honein-AbouHaidar, Lamis Jomaa.

**Writing – original draft:** Dana Nabulsi, Fida Abou Hassan, Gladys Honein-AbouHaidar, Lamis Jomaa.

**Writing – review & editing:** Dana Nabulsi, Hussein Ismail, Fida Abou Hassan, Gladys Honein-AbouHaidar, Lamis Jomaa.

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
