## [Decision Letter · Decision Letter 0]

9 Sep 2020

PONE-D-20-24279

Voices of the Vulnerable: Exploring the livelihood strategies, coping mechanisms and their impact on food insecurity, health and access to health care among Syrian refugees in Lebanon

PLOS ONE

Dear Dr. Jomaa,

Thank you for submitting your manuscript to PLOS ONE. After careful consideration, we feel that it has merit but does not fully meet PLOS ONE’s publication criteria as it currently stands. Therefore, we invite you to submit a revised version of the manuscript that addresses the points raised during the review process.

This is a very well written manuscript. Kindly consider including the study design (mixed methods) in the title. Carefully go through the minor suggestions made by the reviewers, incorporate them and submit it. 

We look forward to receiving your revised manuscript.

Kind regards,

Vijayaprasad Gopichandran

Academic Editor

PLOS ONE

Journal Requirements:

2. Please include additional information regarding the quantitative survey or questionnaire used in the study and ensure that you have provided sufficient details that others could replicate the analyses. For instance, if you developed a questionnaire as part of this study and it is not under a copyright more restrictive than CC-BY, please include a copy, in both the original language and English, as Supporting Information.

3. In the Methods, please discuss whether and how the questionnaire was validated and/or pre-tested. If this did not occur, please provide the rationale for not doing so.

Reviewers' comments:

Reviewer's Responses to Questions

**Comments to the Author**

1. Is the manuscript technically sound, and do the data support the conclusions?

Reviewer #1: Yes

Reviewer #2: Yes

2. Has the statistical analysis been performed appropriately and rigorously? 

Reviewer #1: I Don't Know

Reviewer #2: Yes

3. Have the authors made all data underlying the findings in their manuscript fully available?

Reviewer #1: No

Reviewer #2: Yes

4. Is the manuscript presented in an intelligible fashion and written in standard English?

Reviewer #1: Yes

Reviewer #2: Yes

5. Review Comments to the Author

Reviewer #1: General comments

- The study provides a relevant and useful analysis of the situation and needs of an often hidden population which have not been considered sufficiently in the humanitarian response, and as such provides useful data for the agencies and government. It is a pity that it has taken quite a long time to publish the findings, however the report will still be useful I am sure.

- As the study looked only at the Beqaa valley which the report states represents just over a third of the refugees in Lebanon, the title and conclusions may be more accurate if they were changed to reflect this geographical coverage, rather than all of the Syrian refugees in all of Lebanon.

- Was there any instance in which data triangulation efforts revealed discrepancies, and if so how were these handled?

- Is it possible that the much higher number of female participants (82 women, 40 men) influenced the findings in any way?

- It would be useful to have a copy editor review the document, check for a few instances of unfinished sentences and occasional spelling mistakes (including the correct full name of UNHCR).

Specific comments

Cover page - abstract

Would be useful to mention the gender balance of those interviewed and any gendered implications of the findings.

Would be useful to give a little more explanation of how early marriage is considered an informal livelihood strategy.

Data Availability

Reasons for data not being available are not provided. If the data sets are anonymized in the way described, are there still confidentiality issues?

Table 2

Not sure of the relevance of the p-values here? What do they compare?

Interesting to note the big discrepancy in scores between HFIAS and FCS, particularly amongst those not receiving assistance. Did data triangulation / observation reveal anything about why it would be that food consumption was not so bad, while HFIAS scores were very bad?

p.28 - On the issue of debt, was it considered that being able to access debt is not always negative, but a sign that the people/institutions providing credit have some confidence in getting repaid and therefore a sign of some capacity to pay by the one taking the loans? Otherwise they would not keep increasing credit.

Reviewer #2: This is a very good paper. I included a few suggestions below to help refine it even more:

Abstract:

o Suggestion: take out the details of the methodology from the abstract (leave the highlights only)

Intro:

o Line 55: sentence incomplete

o It would also help in the intro to contextualize the case of Lebanon as a refugee host country (especially historically and considering the disproportionality of refugee v citizens

o Line 66 suggestion to define the meaning and stats of registered v unregistered refugees

o Line 74: what about the xenophobic and anti-refugee sentiments and discrimination that are increasingly reported in Lebanon specifically.

o Line 80: citation for definition of coping? might benefit from grounding coping mechanisms in previous literature defining and discussing the concept in different other contexts. E.g. (Camino et al, (2005). Szczepanikova (2005), Franz (2003), Säävälä (2005).

o It might help at the end of the intro to map the study (e.g. in the next section we will discuss x, then y, then z and will conclude with a,b,c).. also, how such study can benefit us and who would it benefit?

Methods

o Strong methods section – only comment is around the translation: it would be helpful to reflect a bit on any challenges or loss of meaning during this process.

Results

o Any explanation of the disproportion between men and women in the sample?

o apologies if I missed that part: is there a clear distinction in the analysis, in addition to registered and unregistered refugees, between urban refugees (in the city or even rural areas) and those in camps? I think it is an important distinction that might be helpful to highlight more clearly throughout the emerging themes section (it also has an important empirical contribution to understanding the experiences of refugees in encampment v those in the city)

Emergent themes

o Line 340: child labor normalized where? In Syria? Lebanon? Due to displacement?

o Line 350: same thought: child marriage is acceptable among the refugees only? Or did it exist prior to displacement? Did it increase after? Also, a definition is needed for these thorny notions. And what’s the age threshold?

o Line 377: this is very interesting and important and might benefit from being highlighted and grounded a bit more in literature on refugee resilience and coping.

o Line 444: this is a good example to identify any distinction between those in camps v those in the city.

Discussion

o Line 519: what is this assessment? What is its source and what makes it reliable? Also, any citation?

o Line 540-543: very important reflection>> the gap between what’s told and what’s experienced.

Implications and recommendations

o I’m interested in the definition of vulnerability, specifically how this study defines it.

o Here again identifying camp v city refugees in terms of vulnerability assessment can be informative to the humanitarian system

o Line 624: if you haven’t introduced the meaning of protracted displacement you might want to elaborate and also cite the sources.

o The definition of empowerment has proven to be controversial (especially in non-western contexts) so a definition and citation could be helpful here.

o Line 635: Other than a suggestion to integration into workforce, it is not clear what this study is suggesting as an alternative to the current humanitarian system and policies.

Conclusion

o Good summary, however, the contribution of the study and how it benefits different audience, including policy makers, humanitarian organizations, and researcher might need to be spelled out more clearly.

6. PLOS authors have the option to publish the peer review history of their article (what does this mean?). If published, this will include your full peer review and any attached files.

Reviewer #1: No

Reviewer #2: No

---

## [Author Response · Author response to Decision Letter 0]

20 Oct 2020

Please find the detaild point by point response letter submitted along with the revised version of the manuscript in both clean and tracked format. Thank you in advance for your kind consideration.

---

## [Editor Report · Decision Letter 1]

3 Nov 2020

Voices of the Vulnerable: Exploring the livelihood strategies, coping mechanisms and their impact on food insecurity, health and access to health care among Syrian refugees in the Beqaa region of Lebanon

PONE-D-20-24279R1

Dear Dr. Jomaa,

We’re pleased to inform you that your manuscript has been judged scientifically suitable for publication and will be formally accepted for publication once it meets all outstanding technical requirements.

Kind regards,

Vijayaprasad Gopichandran

Academic Editor

PLOS ONE
---

## [Editor Report · Acceptance letter]

9 Nov 2020

PONE-D-20-24279R1 

Voices of the Vulnerable: Exploring the livelihood strategies, coping mechanisms and their impact on food insecurity, health and access to health care among Syrian refugees in the Beqaa region of Lebanon 

Dear Dr. Jomaa:

I'm pleased to inform you that your manuscript has been deemed suitable for publication in PLOS ONE. Congratulations! Your manuscript is now with our production department. 

Kind regards, 

on behalf of

Dr. Vijayaprasad Gopichandran 

Academic Editor

PLOS ONE